## [Transparent Peer Review file · Communications Biology]

Exploring microbial diversity in subsurface aquifers by cell-size fractionated enrichment incubations

Corresponding Author: Mark Dopson

Version 0:

Reviewer comments:

Reviewer #1

(Remarks to the Author)

The manuscript addresses an interesting and important topic by investigating subsurface microbial trophic interactions using a cell-size fractionation approach, offering valuable insights into the survival strategies of bacteria and archaea in low-energy groundwater environments. Overall, the study is well-conducted, and the manuscript is of good quality with solid data and analysis. However, there are some issues that need to be addressed. The novelty and specific objectives of the study are not clearly articulated, which affects the clarity of the research focus. Additionally, the flow and organization of the manuscript could be improved to enhance readability and coherence. It is strongly recommended that the authors carefully review the relevant literature, thoroughly revise the manuscript, and resubmit after making substantial corrections.

Major Comments:

1. The Introduction should explicitly state the research questions and hypotheses. It is currently not clear why cell-size fractionated enrichments provide new insights compared to previous methods. The significance of studying trophic interactions in the terrestrial subsurface must be emphasized with a concise rationale and clear statement of how this work fills a gap in current knowledge. Without a defined hypothesis or objective, the reader may find it difficult to understand the purpose and innovation of the research.
2. The experimental design and methodology require more clarity and detail. The description of the cell-size fractionation protocol and incubation conditions is brief and does not justify key choices (e.g., filter sizes, incubation duration, or media composition). It is recommended to explain why certain size fractions were chosen and how these relate to expected trophic groups. Additionally, the manuscript does not discuss replication or controls (e.g., unfractionated controls or sterile controls), which are essential for validating enrichment results. The logical flow from sample collection to fractionation to analysis should be clearly described so that the rationale for each step is evident.
3. Some paragraphs in the Results and Discussion sections seem disjointed, jumping between topics without clear transitions. For example, the connection between the incubation experiments and the metagenomic dataset is not clearly explained and should be made more explicit to help readers understand how these components relate to each other. In addition, a paragraph might suddenly shift from discussing a microbial group to a separate ecological concept without linking them logically. Group related findings together and ensure each paragraph has a clear topic sentence. The manuscript would benefit from an outline structure (e.g., separate subheadings for each major result or theme) to guide the reader. Additionally, consistent use of terminology is important. Terms like “subsurface”, “enrichment fraction”, or “trophic interaction” should be clearly defined when first introduced and used consistently thereafter.
4. The language and writing style have instances of complex or unclear phrasing that hinder readability. Some sentences are overly long and contain multiple clauses, which makes them difficult to follow. It is recommended to shorten complex sentences and clarify the subject of each sentence. Technical terms should be explained in context; for example, if “trophic interaction” is used, a brief definition of what is meant in this study would help the reader. There are occasional instances of awkward phrasing or minor grammatical issues (see Minor Comments below) that should be addressed. Overall, a thorough proofreading and possibly professional language editing would enhance clarity.
5. Finally, the conclusions need to be more closely tied to the presented data. Currently, the Discussion makes some broad claims about subsurface ecology that extend beyond the direct findings of the experiment. It would strengthen the paper to explicitly link each conclusion to specific results. Moreover, the manuscript would benefit from a clearer articulation of the overall significance of the study and an outlook on future research directions to highlight its contribution to the field.

Minor Comments:

1. Line 26 and elsewhere: Taxonomic units should be italicized.
2. Line 23 and elsewhere: Please change "three groundwaters" to "three types of groundwater".
3. Line 47: The full name of CPR should be mentioned.
4. Line 82-84: Besides 16S rRNA gene sequencing, metagenomic sequencing is also conducted, which is missing here.
5. Line 90-94 and elsewhere: Please ensure that numbers exceeding three digits include commas for readability (e.g., 7,309 instead of 7309).
6. Line 133-136: What could be the potential reason for the absence of Campylobacterota in the marine groundwater?
7. Line 138-140: Has the community composition from previous studies also been included in this study? If not, it would be beneficial to include it.
8. Line 143: Just curious, why did you choose the meteoric groundwater as the initial inoculum?
9. Line 151-154: The phrase "the lack of growth was also reflected in alpha diversity" is unclear and requires clarification.
10. Line 166-168: The rationale for comparing the diversity of Patescibacteria and Nanobdellota at this point in the text should be more clearly explained.
11. Line 177-180: Is the inference regarding these clades' ability to initiate cell division derived from relative abundances in the 16S rRNA gene datasets, or is it supported by qPCR results?
12. Line 183-184: Please clarify the reasoning behind selecting "17 weeks" as a specific time point for analysis.
13. Line 187-189: The relative abundance of ultra-small cells in some saline groundwater is comparable to that in meteoric groundwater.
14. Line 244 and elsewhere: The full name of genes (e.g. *acs*) should be stated or provided in the supplementary tables.
15. Line 450-455: For the network analysis, please specify whether previous data were reanalyzed and integrated with the current dataset.
16. General: Check for consistent use of past vs. present tense. Methods and results are typically past tense (e.g., "we observed"), whereas general statements in the Discussion can be present tense.
17. Please clarify whether the plot represents average values. If so, standard deviations or other measures of variability should also be included.
18. Supplementary Fig. S4: The relative abundance of Patescibacteria is high in some fractionated samples; however, the manuscript states that they did not exhibit growth. This apparent discrepancy should be addressed.
19. Supplementary Fig. S5 and S6: What does the 'sqrt relative abundance' on the y-axis represent exactly? Could you please clarify this term?

Overall, the manuscript addresses an important area of geomicrobiology, but it would benefit from clearer presentation of its novelty, more thorough description of methods, and a tighter link between data and conclusions. The above comments are intended to help improve the clarity and impact of the work. The final decision rests with the editor and other anonymous reviewers, who will assess the article based on their expertise and the journal's criteria.

Reviewer #2

(Remarks to the Author)

In the manuscript "Exploring trophic interactions in the terrestrial subsurface by cell-size fractionated enrichment incubations", Westmejer et al. carry out a sequence-based analysis of microbial communities found in groundwater wells on Åspö island. Using 16S rRNA gene amplicons and metagenomic analysis of "meteoric", "marine" and "saline" groundwater as well as several incubation time-series, they present a taxonomic description of microbial communities and a small number of functional insights on a set of organisms from this system. Using co-occurrence networks, they further argue that they identify potential interactions in the groundwater well ecosystem.

The manuscript provides a descriptive report on a groundwater system that researchers that study such systems, and especially those with an interest in Patescibacteria, would find interesting. Patescibacterial abundance is elevated according to their amplicon analysis on groundwater samples, even though the primers are not particularly suited for the group (<https://microbiomejournal.biomedcentral.com/articles/10.1186/s40168-024-01769-1>). This audience would also be interested in the interactions between Patescibacteria and other bacteria, but the strength of this aspect of the work is oversold in the title and abstract. The bioinformatic methodology (apart from the construction of the interaction network) was strong and the authors used standard, well-accepted pipelines that were suitable for their datasets. However, the data is not organised appropriately to allow reproducibility. It would benefit from increased attention to reporting standards for metadata, clarity around some procedures, and performance indicators of bioinformatic processing. Improving these aspects would increase confidence in the data and results presented, making it more likely that the study will garner interest.

Specific requests (details on each to follow):

- 1) Revise the abstract and title to reflect the dearth of evidence for interactions
- 2) Improve reporting of sample and read set metadata, ensuring that physical samples can be clearly connected to the datasets produced from them
- 3) Include performance indicators for metagenomic assembly and binning
- 4) Provide an ASV count table, including the derived ASV sequence and classifications for each ASV. Ensure that these tables include counts generated for negative controls.
- 5) Clarify methodological procedures around PCR
- 6) Provide a reproducible workflow, including input tables for producing the graphs.

1: The co-occurrence network does not indicate interaction. When data are combined that have been derived from different

physicochemical or spatial environments, co-occurrence better reflects environmental filtering rather than interaction. This is alluded to at the very end of the manuscript and was generally handled well throughout the main text, so we suggest revising the title and abstract to be better aligned with the work as a whole. Also consider using software, such as sparCC, that is regularly used for building networks for example see <https://www.nature.com/articles/s41467-024-49150-y#Sec10>

2: The authors should provide a clear metadata table of each sample that was sequenced and include all metadata available for that sample. The study is quite complex in the number of procedures carried out on different material and it is easy to get lost. In their study, the authors provide data for three sets of physical material: The first is material accumulated on 0.1 µm filters from pumping groundwater. This material was the source of 16S rRNA gene amplicons shown in Figure 1. The second, termed “the initial experiment”, consists of a ten week incubation/time series comparing filtered (<0.45 µm) meteoric groundwater vs unfiltered meteoric groundwater amended with either cell lysate or acetate. This set of samples provided RT-PCR data as well as 16S rRNA gene amplicons that are used as the basis for other main figures. The third set is a similar set of incubations with each of the three types of water, this time over 17 weeks. This set of materials provided cell counts as well as additional 16S rRNA gene amplicon datasets that are largely used for supplemental figures.

The lack of a clear key to the different sample types and inability to connect these physical specimens to the data produced from each leads to a perception of a lack of controls. For instance, there seem to be no incubation control for which cells are not given acetate or cell lysate. These controls would help make better sense of the lack of agreement between the “initial experiment” and the follow-up experiment by defining more clearly a bottle/incubation effect independent of amendment. Furthermore, the authors suggest that negative controls were sequenced for the 16S rRNA amplicons, but those results have not been shown. Inclusion of this information in a metadata table and providing the sequences obtained through the negative controls are essential to provide confidence in the study.

3: To have confidence in the results of their procedures, it is important to document performance indicators. The overall-summary.tsv table, available in the Zenodo repository, nicely summarises the performance of denoising into ASVs but should be included as a supplemental table. It should also be possible to clearly understand which read set corresponds to which sample/experimental condition and to include negative control read sets. A corresponding tracking file for metagenome assemblies would also be helpful. How many reads were in each of the original datasets? Provide quast statistics for each assembly. How many reads mapped back to the assembly they produced?

4: For ASVs, Please also include an ASV table as a supplemental table, showing counts for each ASV across all samples (including negative controls), along with the denoised sequence and the classification. For MAGs, provide a similar table that shows which sample the MAG arose from and how many of the reads of each read set maps back onto each MAG. Do these abundances reinforce the impressions of abundance from the amplicon analyses? This information could be included in the bin_summary.tsv file and included as a supplemental table.

5: There is very little information given as to the PCR conditions, barcoding procedure, and bioinformatic processing necessary to separate samples from one another, details of which are important for interpreting the results.

6: The scripts provided through the github and Zenodo are not sufficient to replicate the work. The script calls data files that are not available. Also, the data that has been deposited at ENA cannot be readily matched with the different sample material and treatment that it arises from. This issue would be remediated if proper metadata reporting standards are met, as outlined above.

Minor comments:

For relative abundance figures, how distinctions and cut-offs for ‘other’ and ‘low abundance’ should be listed in the methods and/or figure legends where they appear.

The naming used should be consistent throughout, for instance, Nanobdellota is used in the manuscript and Supplementary Table S2, while in the Zenodo “bin-summary” document, Nanoarchaeota is used. Similarly, as above, the names of genomes/read files in the bin-summary document, supplementary, and SRA should be consistent.

Lines 390-392 and 397-398 – what were the DNA yields for the incubations? I only see DNA yield numbers for the groundwater in Supplementary table S1

Lines 406-407: Please include more information on the procedure for barcoding and equimolar pooling 16S rRNA gene amplicons. The referenced manuscript (Westmeijer et al, 2024) used 32 cycles of amplification and suggests barcodes were added using a second round of PCR. Please clarify these details and include a table showing DNA concentrations of each sample, procedural blanks, and volume added of each to produce an equimolar pooled library.

Line 413 – Do these primers hit Patescibacteria and other bacteria found in your samples?

Lines 440-441: similar to the request above for amplicons. Provide more information for on the procedure for barcoding and equimolar pooling, including DNA concentrations and volumes used for each sample. Include Run accession IDs from ENA to avoid ambiguity.

Line 445 – how were barcodes handled to ensure data were properly sorted into sequencing libraries?

Lines 459-460 – unclear how Bonferroni is applied to R-squared values.

Line 675 (Figure 4 legend) – it is unclear how abundance was assessed. No abundance information has been provided for any of the MAGs.

Reviewer #4

(Remarks to the Author)

I co-reviewed this manuscript with one of the reviewers who provided the listed reports. This is part of the Communications Biology initiative to facilitate training in peer review and to provide appropriate recognition for Early Career Researchers who co-review manuscripts.

Version 1:

Reviewer comments:

Reviewer #1

(Remarks to the Author)

The authors have done a commendable job addressing the concerns raised in the previous round of review. The manuscript is now better organized, the conclusions are more grounded in the presented results, and the overall flow has improved. I have no further major objections and believe the manuscript makes a valuable contribution to the understanding of subsurface microbial ecology, particularly regarding Patescibacteria and other ultra-small cell lineages.

Reviewer #2

(Remarks to the Author)

I would like to thank the authors for their careful revisions to the manuscript, in particular the inclusion of metadata, performance data, taxonomy data and count data for ASVs was appreciated. The methods are much clearer and the additional information provided allows for a more robust examination of the stated findings.

The data for controls was especially informative. I was somewhat surprised that the authors showed supplemental figure S3 without much explanation, except to argue that the communities differed from the incubations (lines 232-234). Figure S3 shows many of the same phyla that are discussed in regard to the incubations. Due to these similarities, I examined the ASV data supplied on Zenodo to explore the ASVs detected in the controls. I found that many of the abundant ASVs in the controls were identical to ASVs that dominated enrichment communities:

ASV 1ba7fa9612d4779f7da44185aa4c39d5 (Spirochaetota) comprises ~20% of the community in incubation control P21015_1091, which is the dominant ASV (up to 80% rel. abund) recovered from the acetate fractionated incubations from time points 4-10.

ASVs 64158fead87302e45c56df308c54889b and bbf56b54f36dc4b1875f0368c6fd1290 comprise 12-15% of incubation control P21015_1085 and are dominant Acidobacteriota (up to 45% of the relative abundance) of the community found in the cell lysate unfractionated incubations and timepoint 4 of the fractionated cell lysate incubation.

ASV 0c8a863aa50bf14b21e872c802c64c60 comprise up to ~3.5% of controls and is the dominant Pseudomonadota in acetate non-fractionated incubations.

From this, it is unclear if the dominant bacteria detected in enrichments are cellular contamination or DNA contamination. The implications are important for nearly every aspect of the manuscript that relies on ASV abundance data. The qPCR would suggest that it is cellular contamination, but why it would also be found in the extraction controls is somewhat a puzzle. Its presence in the extraction control suggests that it could be DNA contamination. The authors would be doing themselves and their audience a great service by conducting a more thorough investigation on the controls. Do their conclusions hold when the controls are carefully considered?

There was also some information requested but not supplied:

The ASV nucleotide sequences (for being able to evaluate taxonomic classification and chimera detection) mapped read counts for MAGs across metagenomic libraries (read depth is also a MAG quality criterion)

Reviewer #4

(Remarks to the Author)

I co-reviewed this manuscript with one of the reviewers who provided the listed reports. This is part of the Communications Biology initiative to facilitate training in peer review and to provide appropriate recognition for Early Career Researchers who co-review manuscripts.

Version 2:

Reviewer comments:

Reviewer #2

(Remarks to the Author)

I would like to thank the authors for their attention to detail in revising the manuscript. In particular, the inclusion of all relevant data/scripts along with the clear explanation on how negative controls were incorporated provides confidence in their conclusions. I have no further requests.

Reviewer #4

(Remarks to the Author)

I co-reviewed this manuscript with one of the reviewers who provided the listed reports. This is part of the Communications Biology initiative to facilitate training in peer review and to provide appropriate recognition for Early Career Researchers who co-review manuscripts.

Reference: COMMSBIO-25-6919

Title: Exploring microbial diversity in the terrestrial subsurface by cell-size fractionated enrichment incubations

Umeå, 7 November 2025

Dear reviewers,

Thank you very much for the detailed reviews of our manuscript and we hereby submit a substantially revised version of the original manuscript. We compiled the comments below and provided a point-by-point response. All comments were carefully considered, and we provide an annotated version of the revised manuscript to highlight all changes. Please note that the line numbers refer to the revised manuscript.

George Westmeijer et al.

Reviewer #1

The manuscript addresses an interesting and important topic by investigating subsurface microbial trophic interactions using a cell-size fractionation approach, offering valuable insights into the survival strategies of bacteria and archaea in low-energy groundwater environments. Overall, the study is well-conducted, and the manuscript is of good quality with solid data and analysis. However, there are some issues that need to be addressed. The novelty and specific objectives of the study are not clearly articulated, which affects the clarity of the research focus. Additionally, the flow and organization of the manuscript could be improved to enhance readability and coherence. It is strongly recommended that the authors carefully review the relevant literature, thoroughly revise the manuscript, and resubmit after making substantial corrections.

Reply: Thank you for the helpful comments, and we have followed the advice while preparing our revised version. Please note that the line numbers below refer to the revised manuscript.

Major Comments:

1. The Introduction should explicitly state the research questions and hypotheses. It is currently not clear why cell-size fractionated enrichments provide new insights compared to previous methods. The significance of studying trophic interactions in the terrestrial subsurface must be emphasized with a concise rationale and clear statement of how this work fills a gap in current knowledge. Without a defined hypothesis or objective, the reader may find it difficult to understand the purpose and innovation of the research.

Reply: We added our research question and hypothesis to the final paragraph of the introduction: "The primary research question was to explore how co-occurring populations responded to allochthonous organic carbon. We hypothesized that the populations in the fractionated incubations would especially benefit from the cell lysate as it contained a more diverse mixture of organic carbon compounds." (lines 84-87)

2. The experimental design and methodology require more clarity and detail. The description of the cell-size fractionation protocol and incubation conditions is brief and does not justify key choices (e.g., filter sizes, incubation duration, or media composition). It is recommended to explain why certain size fractions were chosen and how these relate to expected trophic groups. Additionally, the manuscript does not discuss replication or controls (e.g., unfractionated controls or sterile controls), which are essential for validating enrichment results. The logical flow from sample collection to fractionation to analysis should be clearly described so that the rationale for each step is evident.

Reply: Thank you for the suggestion. We have added a subsection "Inoculation, incubation, and sampling" that describes the fractionation and incubation conditions in more detail. We also added how the media controls were made plus data on the controls to Fig. 3. We hope by adding this subsection it is easier to understand how the incubations were made. (lines 404-427)

3. Some paragraphs in the Results and Discussion sections seem disjointed, jumping between topics without clear transitions. For example, the connection between the incubation experiments and the metagenomic dataset is not clearly explained and should be made more explicit to help readers understand how these components relate to each other. In addition, a paragraph might suddenly shift from discussing a microbial group to a separate ecological concept without linking them logically. Group related findings together and ensure each paragraph has a clear topic sentence. The manuscript would benefit from an outline structure (e.g., separate subheadings for each major result or theme) to guide the reader. Additionally, consistent use of

terminology is important. Terms like “subsurface”, “enrichment fraction”, or “trophic interaction” should be clearly defined when first introduced and used consistently thereafter.

Reply: We have removed unnecessary jargon and have streamlined the results with a focus on diversity. We have rephrased many paragraphs, especially in the sections "Initial enrichment incubations" and "Incubations with meteoric, marine, and saline groundwater". (e.g., lines 150-233)

4. The language and writing style have instances of complex or unclear phrasing that hinder readability. Some sentences are overly long and contain multiple clauses, which makes them difficult to follow. It is recommended to shorten complex sentences and clarify the subject of each sentence. Technical terms should be explained in context; for example, if “trophic interaction” is used, a brief definition of what is meant in this study would help the reader. There are occasional instances of awkward phrasing or minor grammatical issues (see Minor Comments below) that should be addressed. Overall, a thorough proofreading and possibly professional language editing would enhance clarity.

Reply: Thank you for bringing this to our attention. Throughout the manuscript we have removed unnecessary jargon and have shortened sentences. We have removed the phrase "trophic interaction" and discussed co-occurrences instead.

5. Finally, the conclusions need to be more closely tied to the presented data. Currently, the Discussion makes some broad claims about subsurface ecology that extend beyond the direct findings of the experiment. It would strengthen the paper to explicitly link each conclusion to specific results. Moreover, the manuscript would benefit from a clearer articulation of the overall significance of the study and an outlook on future research directions to highlight its contribution to the field.

Reply: We have rephrased the conclusion to reflect the presented data more. (lines 321-336)

Minor Comments:

1. Line 26 and elsewhere: Taxonomic units should be italicized.

Reply: While we acknowledge journals have differing instructions, we followed the Nature guidelines on taxonomic units to italicize genus and species, but to not italicize higher taxonomic ranks (such as phylum).

2. Line 23 and elsewhere: Please change “three groudwaters” to “three types of groundwater”.

Reply: Done, this sentence now reads: "This study explored potential interactions among bacteria and archaea using anaerobic enrichment incubations with three types of groundwater of contrasting chemistry and hydrological recharge rates." (lines 22-24)

3. Line 47: The full name of CPR should be mentioned.

Reply: Done, this sentence now reads "Patescibacteria (synonym candidate phyla radiation or CPR clade), (..)" (lines 47-48)

4. Line 82-84: Besides 16S rRNA gene sequencing, metagenomic sequencing is also conducted, which is missing here.

Reply: Thank you for noticing this. This sentence has been clarified and now reads: "The microbial communities in the incubations were first characterized using 16S rRNA gene sequencing, followed by metagenomic sequencing of selected cultures based on taxonomic composition." (lines 87-89)

5. Line 90-94 and elsewhere: Please ensure that numbers exceeding three digits include commas for readability (e.g., 7,309 instead of 7309).

Reply: This has been changed throughout the manuscript.

6. Line 133-136: What could be the potential reason for the absence of Campylobacterota in the marine groundwater?

Reply: This could be related to hydrochemistry. We phrased it carefully as: "(..) while this phylum was not detected in the marine groundwater that could be due to the lower concentration of reduced sulfur species (H₂S and HS⁻) in the latter groundwater type (Table 1)." (lines 141-143)

7. Line 138-140: Has the community composition from previous studies also been included in this study? If not, it would be beneficial to include it.

Reply: We considered including the data from previous studies in Fig. 1 but chose not to as we already included the data for the co-occurrence analysis and preferred not to use the data twice.

8. Line 143: Just curious, why did you choose the meteoric groundwater as the initial inoculum?

Reply: The meteoric groundwater was chosen as it has relatively high cell numbers compared to the other two groundwater types and a high diversity of clades with potentially associated lifestyles.

9. Line 151-154: The phrase "the lack of growth was also reflected in alpha diversity" is unclear and requires clarification.

Reply: Agreed, this sentence now reads: "Fractionated incubations had a significantly higher alpha diversity compared to the non-fractionated incubations, despite having received part of the inoculum (Shannon's H 3.6 and 1.9, respectively; Mann-Whitney test *p*-value 0.02, *n* = 40)." (lines 158-160)

10. Line 166-168: The rationale for comparing the diversity of Patescibacteria and Nanobdellota at this point in the text should be more clearly explained.

Reply: Thank you for the suggestion. We clarified this sentence and it now states: "Among these, the Patescibacteria displayed the highest diversity, comprising 20 classes and 97 orders. In comparison, the Nanobdellota contained

only a single class and seven orders, underscoring that a substantial proportion of the microbial diversity in the fractionated cultures was affiliated with the Patescibacteria." (lines 179-182)

11. Line 177-180: Is the inference regarding these clades' ability to initiate cell division derived from relative abundances in the 16S rRNA gene datasets, or is it supported by qPCR results?

Reply: This was supported by the qPCR results showing an increase in gene copy numbers, combined with the 16S rRNA gene data. This was clarified and now reads: "In general, microbial abundance (both qPCR and epifluorescence microscopy) and 16S rRNA gene amplicon sequencing showed that the non-fractionated incubations were strongly enriched in either Bacillota, Spirochaetota, or Desulfobacterota. These non-fractionated incubations differentiated over time as the cell abundance increased (approximately ten-fold) and the diversity decreased, suggesting that populations affiliated with these phyla were capable of cell division (Supplemental Fig. S5). " (lines 223-227)

12. Line 183-184: Please clarify the reasoning behind selecting "17 weeks" as a specific time point for analysis.

Reply: This had a practical reason, it being an extensive field work campaign that prevented us taking further measurements and extending the experiment to e.g., 20 weeks.

13. Line 187-189: The relative abundance of ultra-small cells in some saline groundwater is comparable to that in meteoric groundwater.

Reply: Correct, this sentence has been modified and now reads: "Size fractionation of the saline groundwater failed as no cells were detected in the incubations, possibly due to a lower cell number in this groundwater combined with a slightly lower abundance of ultra-small cells in this groundwater (Fig. 1)." (lines 208-210)

14. Line 244 and elsewhere: The full name of genes (e.g. acs) should be stated or provided in the supplementary tables.

Reply: Supplementary Table S3 now includes the abbreviation of the genes and the full names.

15. Line 450-455: For the network analysis, please specify whether previous data were reanalyzed and integrated with the current dataset.

Reply: Thank you for noticing this. A sentence has been added about the integration of previously published data: "To explore potential host-symbiont relationships, a co-occurrence analysis based on ASVs was performed using both previously published data on 24 groundwaters intersected by Äspö HRL ($n = 72$)¹¹, combined with the environmental data generated within this study ($n = 18$) and the non-fractionated incubations ($n = 50$). " (lines 293-296)

16. General: Check for consistent use of past vs. present tense. Methods and results are typically past tense (e.g., "we observed"), whereas general statements in the Discussion can be present tense.

Reply: Thank you. We found some inconsistencies mainly in the methods and corrected these.

17. Please clarify whether the plot represents average values. If so, standard deviations or other measures of variability should also be included.

Reply: We included the standard deviation in Fig. 2 as a vertical line, but this was mainly hidden behind the points. So, we doubled the standard deviation and slightly reduced the size of the points to make the variation visible to the reader. We described this in the caption as: "Points represent mean values and the vertical bars indicate ± 2 standard deviations." (lines 708-709)

18. Supplementary Fig. S4: The relative abundance of Patescibacteria is high in some fractionated samples; however, the manuscript states that they did not exhibit growth. This apparent discrepancy should be addressed.

Reply: We clarified this statement by writing: "Patescibacteria, Nanobdellota, and Omnitrophota were dominant in these incubations and responsible for up to 90 % of the sequence reads, after being enriched with a 0.45 μm filtration of the inoculum. Despite the high diversity of populations affiliated with these phyla, these populations did not demonstrate any increase in cell numbers during the incubation period, despite the amendment of single (acetate) and more complex (lysate) carbon sources." (lines 229-233)

19. Supplementary Fig. S5 and S6: What does the 'sqrt relative abundance' on the y-axis represent exactly? Could you please clarify this term?

Reply: This has been clarified in the captions of the now figures S6 and S7: "The black line represents a linear fit among the square root transformed relative abundances of the Patescibacteria (x) and the phylum of interest (y) in each sample."

Overall, the manuscript addresses an important area of geomicrobiology, but it would benefit from clearer presentation of its novelty, more thorough description of methods, and a tighter link between data and conclusions. The above comments are intended to help improve the clarity and impact of the work. The final decision rests with the editor and other anonymous reviewers, who will assess the article based on their expertise and the journal's criteria.

Reply: We are grateful for the comprehensive review and hope to have adequately met your comments.

Reviewer #2 (Remarks to the Author):

In the manuscript "Exploring trophic interactions in the terrestrial subsurface by cell-size fractionated enrichment incubations", Westmeijer et al. carry out a sequence-based analysis of microbial communities found in groundwater wells on Åspö island. Using 16S rRNA gene amplicons and metagenomic analysis of “meteoric”, “marine” and “saline” groundwater as well as several incubation time-series, they present a taxonomic description of microbial communities and a small number of functional insights on a set of organisms from this system. Using co-occurrence networks, they further argue that they identify potential interactions in the groundwater well ecosystem.

The manuscript provides a descriptive report on a groundwater system that researchers that study such systems, and especially those with an interest in Patescibacteria, would find interesting. Patescibacterial abundance is elevated according to their amplicon analysis on groundwater samples, even though the primers are not particularly suited for the group (<https://microbiomejournal.biomedcentral.com/articles/10.1186/s40168-024-01769-1>). This audience would also be interested in the interactions between Patescibacteria and other bacteria, but the strength of this aspect of the work is oversold in the title and abstract. The bioinformatic methodology (apart from the construction of the interaction network) was strong and the authors used standard, well-accepted pipelines that were suitable for their datasets. However, the data is not organised appropriately to allow reproducibility. It would benefit from increased attention to reporting standards for metadata, clarity around some procedures, and performance indicators of bioinformatic processing. Improving these aspects would increase confidence in the data and results presented, making it more likely that the study will garner interest.

Specific requests (details on each to follow):

- 1) Revise the abstract and title to reflect the dearth of evidence for interactions
- 2) Improve reporting of sample and read set metadata, ensuring that physical samples can be clearly connected to the datasets produced from them
- 3) Include performance indicators for metagenomic assembly and binning
- 4) Provide an ASV count table, including the derived ASV sequence and classifications for each ASV. Ensure that these tables include counts generated for negative controls.
- 5) Clarify methodological procedures around PCR
- 6) Provide a reproducible workflow, including input tables for producing the graphs.

1: The co-occurrence network does not indicate interaction. When data are combined that have been derived from different physicochemical or spatial environments, co-occurrence better reflects environmental filtering rather than interaction. This is alluded to at the very end of the manuscript and was generally handled well throughout the main text, so we suggest revising the title and abstract to be better aligned with the work as a whole. Also consider using software, such as sparCC, that is regularly used for building networks for example see <https://www.nature.com/articles/s41467-024-49150-y#Sec10>

Reply: We have modified the title and the abstract to align it better with the results. The title now reads: "Exploring diversity patterns in the terrestrial subsurface by cell-size fractionated enrichment incubations ". The abstract is on lines 21-32.

2: The authors should provide a clear metadata table of each sample that was sequenced and include all metadata available for that sample. The study is quite complex in the number of procedures carried out on

different material and it is easy to get lost. In their study, the authors provide data for three sets of physical material: The first is material accumulated on 0.1 µm filters from pumping groundwater. This material was the source of 16S rRNA gene amplicons shown in Figure 1. The second, termed “the initial experiment”, consists of a ten-week incubation/time series comparing filtered (<0.45 µm) meteoric groundwater vs unfiltered meteoric groundwater amended with either cell lysate or acetate. This set of samples provided RT-PCR data as well as 16S rRNA gene amplicons that are used as the basis for other main figures. The third set is a similar set of incubations with each of the three types of water, this time over 17 weeks. This set of materials provided cell counts as well as additional 16S rRNA gene amplicon datasets that are largely used for supplemental figures.

The lack of a clear key to the different sample types and inability to connect these physical specimens to the data produced from each leads to a perception of a lack of controls. For instance, there seem to be no incubation control for which cells are not given acetate or cell lysate. These controls would help make better sense of the lack of agreement between the “initial experiment” and the follow-up experiment by defining more clearly a bottle/incubation effect independent of amendment. Furthermore, the authors suggest that negative controls were sequenced for the 16S rRNA amplicons, but those results have not been shown. Inclusion of this information in a metadata table and providing the sequences obtained through the negative controls are essential to provide confidence in the study.

Reply: During the initial experiment two media controls were included, that contained sterile ultrapure water instead of inoculum. These controls are now also included in the ordination plot of the initial experiment (Fig. 3). This ordination now shows how the controls differ from the incubations. The media controls are discussed on lines 163-165, and the community composition is shown along the extraction controls in Supplemental Figure S3. In addition to Supplemental Tables S1, S5, and S6 we also provided the file 'metatdata.tsv' in the Zenodo repository.

3: To have confidence in the results of their procedures, it is important to document performance indicators. The overall-summary.tsv table, available in the Zenodo repository, nicely summarises the performance of denoising into ASVs but should be included as a supplemental table. It should also be possible to clearly understand which read set corresponds to which sample/experimental condition and to include negative control read sets. A corresponding tracking file for metagenome assemblies would also be helpful. How many reads were in each of the original datasets? Provide quast statistics for each assembly. How many reads mapped back to the assembly they produced?

Reply: We have included Supplementary Tables S5 and S6 to match the DNA library with the incubation it was extracted from. Similarly, for the metagenomic sequencing we added Supplementary Table S7 to clarify which culture was used to produce the metagenomes. Accession numbers were also included in this table as was suggested, to enable to identify the data deposited at ENA. The Quast statistics were included as Supplementary data. The file 'overall-summary.tsv' was partly included now as Supplementary Table S5.

4: For ASVs, please also include an ASV table as a supplemental table, showing counts for each ASV across all samples (including negative controls), along with the denoised sequence and the classification. For MAGs, provide a similar table that shows which sample the MAG arose from and how many of the reads of each read set maps back onto each MAG. Do these abundances reinforce the impressions of abundance from the amplicon analyses? This information could be included in the bin_summary.tsv file and included as a supplemental table.

Reply: We chose to make the ASV table ('ASV_table.tsv.gz') available via the GitHub, along with the taxonomy of the ASVs ('ASV_tax.tsv.gz'). This table would be too long to include as a supplemental table (over 70 thousand rows). After doing the bioinformatic processing of the metagenomes, we verified if the genus of the reconstructed genome was present among the amplicon data of the incubations it was reconstructed from. This was intended as a quality check of the library preparation of both data types (amplicons and metagenomes).

5: There is very little information given as to the PCR conditions, barcoding procedure, and bioinformatic processing necessary to separate samples from one another, details of which are important for interpreting the results.

Reply: We clarified the molecular work by adding details regarding PCR conditions and pooling strategy (lines 431-447). We also added two supplementary tables (S5 and S6) showing the DNA concentration of each incubation. Supplementary Table S7 includes details on the metagenomic sequencing and details on the library preparation such as the number of PCR cycles and the incubation the metagenome originated from.

6: The scripts provided through the github and Zenodo are not sufficient to replicate the work. The script calls data files that are not available. Also, the data that has been deposited at ENA cannot be readily matched with the different sample material and treatment that it arises from. This issue would be remediated if proper metadata reporting standards are met, as outlined above.

Reply: We have now added the data that is required to run the scripts to the GitHub repository, including metadata and hydrochemistry. As mentioned above, we added run accessions to Supplementary Table S7 to match the data deposited at ENA with the DNA library and the incubation. We added Supplementary Table S5 and S6 to clarify which incubation was used to generate the DNA library.

Minor comments:

For relative abundance figures, how distinctions and cut-offs for 'other' and 'low abundance' should be listed in the methods and/or figure legends where they appear.

Reply: We have clarified this e.g., in Fig. 1 the ten most abundant phyla are shown: "Community composition on the level of phylum, including the ten most abundant phyla (based on summed relative abundance) while grouping remaining phyla as "Other" and including ASVs without an assigned phylum as "Unidentified"." (lines 697-699)

The naming used should be consistent throughout, for instance, Nanobdellota is used in the manuscript and Supplementary Table S2, while in the Zenodo "bin-summary" document, Nanoarchaeota is used. Similarly, as above, the names of genomes/read files in the bin-summary document, supplementary, and SRA should be consistent.

Reply: We have corrected this inconsistency in the supplementary table (now Supplementary Table S2).

Lines 390-392 and 397-398 – what were the DNA yields for the incubations? I only see DNA yield numbers for the groundwater in Supplementary table S1

Reply: We have included the DNA yields for the incubations as Supplementary Tables S5 and S6.

Lines 406-407: Please include more information on the procedure for barcoding and equimolar pooling 16S rRNA gene amplicons. The referenced manuscript (Westmeijer et al, 2024) used 32 cycles of amplification and suggests barcodes were added using a second round of PCR. Please clarify these details and include a table showing DNA concentrations of each sample, procedural blanks, and volume added of each to produce an equimolar pooled library.

Reply: We provided more details on the molecular work related to the DNA extractions, PCR design, and pooling strategy (lines 430-454). We also included DNA concentration of each library (both for the groundwaters and the incubations) in the Supplementary Tables S1, S5, and S6.

Line 413 – Do these primers hit Patescibacteria and other bacteria found in your samples?

Reply: The 341F and 805R primer pair has been frequently used in the subsurface groundwaters of Äspö HRL as they cover the bacterial domain, including Patescibacteria. Comparing the metagenomic sequencing data with the primer-based assays showed that all major groups (e.g., phyla) of the genome-resolved metagenomes were represented among the ASVs.

Lines 440-441: similar to the request above for amplicons. Provide more information on the procedure for barcoding and equimolar pooling, including DNA concentrations and volumes used for each sample. Include Run accession IDs from ENA to avoid ambiguity.

Reply: A table (Supplementary Table S7) has been added that includes run accessions, DNA concentration, and details regarding the library preparation.

Line 445 – how were barcodes handled to ensure data were properly sorted into sequencing libraries?

Reply: Prior to the lab work, a spreadsheet was made containing the barcodes for each DNA extract and the position of the future library in the cartridge (from the Tecan MagigPrep). The libraries were prepared in batches of six and the tubes containing the DNA extract were aligned with the cartridge. The cartridge itself was also labelled with the library id. Demultiplexing of the sequencing data was done by the sequencing facility.

Lines 459-460 – unclear how Bonferroni is applied to R-squared values.

Reply: Thank you for noticing this inconsistency. We apologize for this error as no Bonferroni correction was applied to the R-squared values as was initially stated in the method section. (lines 505-506)

Line 675 (Figure 4 legend) – it is unclear how abundance was assessed. No abundance information has been provided for any of the MAGs.

Reply: We clarified this inconsistency by instead writing: "Each point represents a MAG ($n = 35$), colored according to phylum while grouping low-abundant phyla (based on 16S rRNA gene amplicons from the incubations) as "Other"." (lines 723-725)

Reviewer #3 (Remarks to the Author):

I co-reviewed this manuscript with one of the reviewers who provided the listed reports. This is part of the Communications Biology initiative to facilitate training in peer review and to provide appropriate recognition for Early Career Researchers who co-review manuscripts.

Reply: We thank the reviewer for taking the time to review our manuscript.

Reference: COMMSBIO-25-6919

Title: Exploring microbial diversity in the terrestrial subsurface by cell-size fractionated enrichment incubations

Umeå, 17 December 2025

Dear reviewer,

Thank you for further investigating our control samples. We also wish to apologize for not having noticed this at an earlier stage. We took these comments very seriously and did a thorough revision of the data analysis. Hence, we have removed the ASVs from the controls from the respective samples and discussed this in the methods, results, figures, supplemental figures, and supplemental tables. We provide an annotated version of the revised manuscript to highlight all recent changes. Please note that the line numbers refer to the revised manuscript.

George Westmeijer et al.

Reviewer #2 (Remarks to the Author):

I would like to thank the authors for their careful revisions to the manuscript, in particular the inclusion of metadata, performance data, taxonomy data and count data for ASVs was appreciated. The methods are much clearer and the additional information provided allows for a more robust examination of the stated findings.

The data for controls was especially informative. I was somewhat surprised that the authors showed supplemental figure S3 without much explanation, except to argue that the communities differed from the incubations (lines 232-234). Figure S3 shows many of the same phyla that are discussed in regard to the incubations. Due to these similarities, I examined the ASV data supplied on Zenodo to explore the ASVs detected in the controls. I found that many of the abundant ASVs in the controls were identical to ASVs that dominated enrichment communities:

ASV 1ba7fa9612d4779f7da44185aa4c39d5 (Spirochaetota) comprises ~20% of the community in incubation control P21015_1091, which is the dominant ASV (up to 80% rel. abund) recovered from the acetate fractionated incubations from time points 4-10.

ASVs 64158fead87302e45c56df308c54889b and bbf56b54f36dc4b1875f0368c6fd1290 comprise 12-15% of incubation control P21015_1085 and are dominant Acidobacteriota (up to 45% of the relative abundance) of the community found in the cell lysate unfractionated incubations and timepoint 4 of the fractionated cell lysate incubation.

ASV 0c8a863aa50bf14b21e872c802c64c60 comprise up to ~3.5% of controls and is the dominant Pseudomonadota in acetate non-fractionated incubations.

From this, it is unclear if the dominant bacteria detected in enrichments are cellular contamination or DNA contamination. The implications are important for nearly every aspect of the manuscript that relies on ASV abundance data. The qPCR would suggest that it is cellular contamination, but why it would also be found in the extraction controls is somewhat a puzzle. Its presence in the extraction control suggests that it could be DNA contamination. The authors would be doing themselves and their audience a great service by conducting a more thorough investigation on the controls. Do their conclusions hold when the controls are carefully considered?

Reply: Thank you for bringing this to our attention. We did a thorough investigation and removed the ASVs in the controls from the respective samples. Although the relative abundances of some clades slightly varied (please see the changes in the annotated manuscript), this did not affect the conclusions nor the general interpretation of the results. Below we clarify how we removed the ASVs and how we described this in the manuscript.

We added a sentence to the methods (subsection 'Bioinformatics and data analysis') to describe how we removed the ASVs: 'ASVs present in the DNA extraction controls of the groundwater sampling and the incubations were removed from the respective samples prior to the analysis. In addition, ASVs from the acetate and lysate controls (four controls in total) were removed from the incubation samples that were enriched with these media.' (lines 501-503).

All the figures, tables, and supplementary figures and tables were remade after removing the contaminant ASVs. We also added a column to supplementary tables S1, S5, and S6 to clarify how many ASVs were removed from each sample. As some ASVs present in the controls were abundant in certain enrichment cultures, we clarified this in the results: 'ASVs detected in the extraction control (358 ASVs) and in the acetate (141 and 183 ASVs) and lysate (233 and 220 ASVs) media controls were removed prior to the analyses. Certain ASVs from the controls affiliated with the Spirochaetota and the Acidobacteriota were abundant in the fractionated acetate incubations and the non-

fractionated lysate incubations, respectively. Although this affected the community structure of these incubations, this did not alter the overall interpretations of the results.' (lines 152-156). A similar sentence was also added to the results of the groundwater community (subsection 'Characterization of the groundwaters used as inoculum'): 'The extraction control for the groundwater sampling contained 376 amplicon sequencing variants (ASVs; Supplemental Fig. S3) that were removed from these samples prior to the analyses (Supplemental Table S1).' (lines 93-95). We also clarified in the caption of Supplemental Fig. S3 that the ASVs were removed from all analysis.

There was also some information requested but not supplied:

**The ASV nucleotide sequences (for being able to evaluate taxonomic classification and chimera detection)
mapped read counts for MAGs across metagenomic libraries (read depth is also a MAG quality criterion)**

Reply: The sequence of the ASVs was added to the Zenodo repository as 'ASV_tax_seqs.tsv.gz'. The reads were mapped to the MAGs using CoverM, and the output was also added to the Zenodo repo ('data/mag/coverm-mapping.tsv').